# 2D QUANTIZATION FOR ULTRA-LOW-BIT OPTIMIZERS

## ABSTRACT

Optimizer states used to accelerate neural network training become a significant memory bottleneck as model size grows. A common mitigation is to compress these high-precision states to low-bit representations, but existing methods typically stop at 4 bits. In this paper, we push the bitwidth of AdamW and Adafactor states down to 1.5 and 2 bits by mapping high-precision values to their nearest low-bit representations in a two-dimensional (2D) polar space, which we call 2D quantization. This is effective because optimizer states exhibit a quasi-Gaussian distribution with strong circular symmetry. To further improve efficiency, we offer concrete design principles for both signed and unsigned data, and we validate the superiority of our approach over traditional 1D quantization through static experiments on real momentum matrices. Across a range of pretraining and fine-tuning benchmarks—including image classification and natural language modeling—our ultra-low-bit AdamW and Adafactor match the performance of their 16/32-bit counterparts while dramatically reducing memory usage.

## 1 INTRODUCTION

Deep neural networks, such as Transformer-based architectures (Vaswani et al., 2017; Liu et al., 2019; Radford et al., 2019; Zhang et al., 2022; Touvron et al., 2023; Yang et al., 2024), have achieved remarkable success in applications such as dialogue systems and multilingual translation. A key enabler of this progress is the AdamW optimizer (Loshchilov & Hutter, 2019), which maintains first- and second-moment states for stable optimization. However, as models scale, the memory required to store these optimizer states becomes a dominant factor in overall training cost.

To reduce the memory consumed by the optimizer states, quantization is a classical and effective strategy. Formally, given a codebook $Q \in \mathbb{R}^k$, any point in $\mathbb{R}^k$ can be approximated by its nearest neighbor in $Q$. This process is known as $k$-dimensional ($k$D) quantization. Prior work on low-bit optimizers (Dettmers et al., 2022; Li et al., 2023) has focused exclusively on 1D quantization, namely quantizing a single high-precision floating-point number into a low-precision number, e.g., converting a 32-bit floating-point number to a 4-bit one.

While 1D quantization is straightforward, its performance degrades dramatically in the ultra-low-bit regime (i.e., less than 4 bits). Its core limitation, as illustrated in Fig. 1(a), is the use of a simple rectilinear grid that quantizes each dimension independently, thus failing to exploit the strong inter-dimensional correlations found in optimizer states. Adopting 2D quantization provides a natural solution by jointly encoding parameters, thereby capturing the underlying data geometry.

This conceptual difference is visualized in Fig. 1. The grid-like structure of a 1D quantizer (Fig. 1(a)) is fundamentally mismatched with the underlying data geometry. As we empirically demonstrate in Appendix D, optimizer states consistently exhibit a quasi-Gaussian distribution with strong circular symmetry. Although a general-purpose 2D quantizer (Fig. 1(b)) can form ideal, data-aware decision cells, it incurs prohibitive storage and computational overheads, rendering it impractical. This motivates our approach: a structured 2D quantizer that is both geometrically congruent and practically efficient. We propose a polar quantizer (Fig. 1(c)), whose concentric codebook structure is explicitly designed to match the iso-probability contours of the source. This design is powerful because it offers theoretical elegance through its structural congruence, enables intelligent, radius-dependent bit allocation, and maintains practical efficiency with a training-free design and fast encoding.

This paper pushes the bitwidth of the moments of Adafactor/AdamW down to 1.5 and 2.0 bits via 2D quantization. Our main contributions are highlighted below.

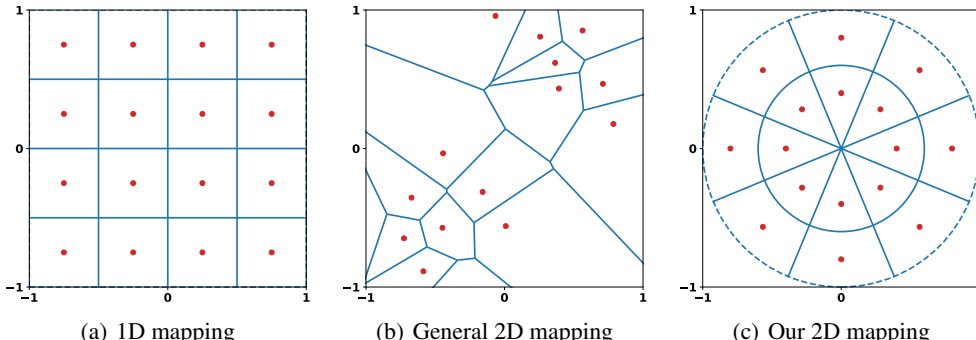

(a) 1D mapping      (b) General 2D mapping      (c) Our 2D mapping

Figure 1: Visualization of 2D quantization mappings. Red dots are codebook vectors and blue lines are decision boundaries. (a) The 1D mapping's rectilinear grid ignores the data's circular symmetry. (b) A general 2D mapping adapts perfectly but is computationally expensive. (c) Our proposed polar mapping offers an efficient, structured design that is congruent with the quasi-Gaussian data distribution.

First, to address the significant performance degradation of traditional 1D quantizers at ultra-low bitwidths (e.g., 2-bit), we propose a 2D quantization framework in the polar coordinate system. This approach is highly effective as it leverages the quasi-Gaussian distribution and strong circular symmetry observed in optimizer states. It enables the creation of ultra-low-bit AdamW/Adafactor optimizers that achieve performance on par with their 16/32-bit counterparts. We validate the superiority of our 2D approach over 1D quantization through static experiments on real-world momentum matrices.

Second, we offer concrete design principles for both signed and unsigned data for moments in optimizer states. For signed inputs, we show that the zero point is unnecessary, that placing codebook magnitudes near the median is beneficial, and we theoretically justify the use of angularly uniform sampling. For unsigned inputs, we propose a novel mapping only in the first quadrant, which enhances efficiency by assigning more points to larger values for precision and fewer to smaller ones. This is combined with small axis offsets to guarantee training stability by preventing division-by-zero errors.

Finally, we evaluate our 1.5-bit and 2-bit AdamW/Adafactor across both language modeling and vision tasks. For language modeling tasks, we pretrain GPT-2 on OpenWebText and LLaMA-2 on C4, and fine-tune LLaMA-2-7B and Qwen2.5-7B on GLUE. For vision tasks, we pretrain ViT-Base and ResNet-50 on ImageNet-1K. Across all these benchmarks, our ultra-low-bit AdamW/Adafactor achieve performance comparable to that of their 16/32-bit counterparts with significantly less memory.

## 2 RELATED WORK

To reduce the memory cost of optimizers, several directions are explored, and are introduced below.

**Low-rank approximation.** Adafactor (Shazeer & Stern, 2018) approximates Adam's second moments by the outer product of two vectors. Feinberg et al. (2023) and Yen et al. (2023) approximate the preconditioner in second-order optimizers via truncated SVD. GaLore (Zhao et al., 2024) projects gradient matrices into a low-rank subspace for memory saving, while Q-GaLore (Zhang et al., 2024) further reduces memory by quantizing the projectors to 4-bit and weights to 8-bit. LoQT (Loeschcke et al., 2024) extends GaLore into QLoRA (Dettmers et al., 2023), enabling LLM pretraining with a 4-bit full-rank matrix and additional low-rank matrices per linear layer.

**Division.** SM3 (Anil et al., 2019) approximates Adam's second moment using its cover statistics. Adam-mini (Zhang et al., 2025) partitions parameters into blocks corresponding to small dense Hessian sub-blocks, allowing shared second moments within each block.

**Quantization.** Dettmers et al. (2022) employ block-wise dynamic quantization to store first-order optimizer states in 8-bit. Li et al. (2023) address the zero-point issue when quantizing Adam/AdamW second moments to 4-bit via a zero-point-free mapping. Wang et al. (2024) show that quantizing the eigenvectors of 4-bit Shampoo outperforms quantizing the preconditioner directly, and Li et al. (2025)

introduce Cholesky quantization for 4-bit Shampoo. All prior methods use 1D quantizers; Tian et al. (2025) proposes a 2D mapping from $\mathbb{R}$ to a closed disk in $\mathbb{R}^2$, allowing fast 2D quantization with linear complexity independent of precision. However, due to theoretical and implementation limits, this only reduces AdamW bitwidth to 3.32 bits for fine-tuning and is ineffective for pretraining.

## 3    METHODOLOGY

**Notations.** We use a non-bold letter like $a$ or $A$ to denote a scalar, a boldfaced lower-case letter like $\boldsymbol{a}$ to denote a vector, and a boldfaced upper-case letter such as $\boldsymbol{A}$ to denote a matrix. $\boldsymbol{x} = [x_i]$ means that the $i$-th element of column vector $\boldsymbol{x}$ is $x_i$ and $\boldsymbol{X} = [\boldsymbol{x}_i]$ means the $i$-th column of matrix $\boldsymbol{X}$ is $\boldsymbol{x}_i$. $\|\boldsymbol{x}\|_p$ denotes the $p$-norm of vector $\boldsymbol{x}$. Given two matrices $\boldsymbol{A}$ and $\boldsymbol{B}$, $\boldsymbol{A} \odot \boldsymbol{B}$ represents the elementwise matrix product (Hadamard product), and $\langle \boldsymbol{A}, \boldsymbol{B} \rangle$ represents the inner product. The Frobenius norm of a matrix $\boldsymbol{A}$ is $\|\boldsymbol{A}\|_F = \sqrt{\langle \boldsymbol{A}, \boldsymbol{A} \rangle}$.

### 3.1    OUR LOW-BIT OPTIMIZATION FRAMEWORK

Low-bit optimizers store optimizer states like first and second moments using low-precision floating-point numbers, temporarily dequantizing them to high precision during computations. This significantly reduces the static memory footprint of these optimizers. Below, we use two widely adopted optimizers AdamW (Loshchilov & Hutter, 2019) and Adafactor (Shazeer & Stern, 2018) as examples.

**Quantized AdamW.** At the $t$-th iteration, given the minibatch gradient $\boldsymbol{g}_t$, we first dequantize the previous first and second moments $\boldsymbol{m}_{t-1}^q$

---

**Algorithm 1** Quantized AdamW

**Input:** Step number $T$, learning rate $\eta$, hyper-parameters $\beta_1, \beta_2$, decay parameter $\lambda, \epsilon$
**Initialize:** $\boldsymbol{\theta}_0, \boldsymbol{m}_0^q \leftarrow 0, \boldsymbol{v}_0^q \leftarrow 0$
1: **for** $t = 1, \ldots, T$ **do**
2:     $\boldsymbol{g}_t \leftarrow \nabla_{\boldsymbol{\theta}} f(\boldsymbol{\theta}_{t-1})$
3:     Dequantize $\boldsymbol{m}_{t-1}, \boldsymbol{v}_{t-1}$ using Eqn. (1)
4:     Update moments $\boldsymbol{m}_t, \boldsymbol{v}_t$ using Eqn. (3)
5:     Update parameters $\boldsymbol{\theta}_t$ using Eqn. (4)
6:     Quantize $\boldsymbol{m}_t^q, \boldsymbol{v}_t^q$ using Eqn. (2)
**Output:** $\boldsymbol{\theta}_T$

---

and $\boldsymbol{v}_{t-1}^q$ via our low-bit dequantizer $\mathcal{D}$ which is introduced in Sec. 3.2:

$$\boldsymbol{m}_{t-1} = \mathcal{D}(\boldsymbol{m}_{t-1}^q), \quad \boldsymbol{v}_{t-1} = \mathcal{D}(\boldsymbol{v}_{t-1}^q). \tag{1}$$

Here $\boldsymbol{m}_{t-1}^q$ and $\boldsymbol{v}_{t-1}^q$ are respectively the low-precision versions of high-precision $\boldsymbol{m}_{t-1}$ and $\boldsymbol{v}_{t-1}$, and are computed by our proposed low-bit quantizer $\mathcal{Q}$ presented in Sec. 3.2:

$$\boldsymbol{m}_{t-1}^q = \mathcal{Q}(\boldsymbol{m}_{t-1}), \quad \boldsymbol{v}_{t-1}^q = \mathcal{Q}(\boldsymbol{v}_{t-1}). \tag{2}$$

In this way, we can follow vanilla AdamW to update the first and second moments $\boldsymbol{m}_t$ and $\boldsymbol{v}_t$:

$$\boldsymbol{m}_t = \beta_1 \boldsymbol{m}_{t-1} + (1 - \beta_1)\boldsymbol{g}_t, \quad \boldsymbol{v}_t = \beta_2 \boldsymbol{v}_{t-1} + (1 - \beta_2)\boldsymbol{g}_t^2. \tag{3}$$

Following (Dettmers et al., 2022; Zhao et al., 2024), we quantize the 2-dimensional optimizer states except those used for updating the embedding layers during transformer-based model training. Besides, since the norm of a tensor can vary significantly after quantization, we scale the learning rate $\eta$ used to update the corresponding trainable tensor, by a scale factor $\alpha$. Accordingly, we update the model parameters via

$$\boldsymbol{\theta}_t = \boldsymbol{\theta}_{t-1} - \eta_t \left( \frac{\alpha \cdot \boldsymbol{m}_t/(1 - \beta_1^t)}{\sqrt{\boldsymbol{v}_t/(1 - \beta_2^t)} + \epsilon} + \lambda \boldsymbol{\theta}_{t-1} \right), \tag{4}$$

where $\eta$ is the learning rate and $\lambda$ is the hyper-parameter for weight decay. Crucially, for the non-quantized layers (i.e., the embedding layers), we set $\alpha = 1.0$, effectively applying the standard update rule. Finally, we quantize the high-precision first and second moments $\boldsymbol{m}_t$ and $\boldsymbol{v}_t$ into low-precision versions $\boldsymbol{m}_t$ and $\boldsymbol{v}_t$ via Eqn. (2).

For clarity, we summarize all steps of quantized AdamW into Algorithm 1, where we use a gray color to highlight the extra steps. One can observe that quantized AdamW almost follows vanilla AdamW and is quite simple. Moreover, its memory cost becomes much lower than AdamW, since it only need to maintain the low-bit first and second moments without performance degradation.

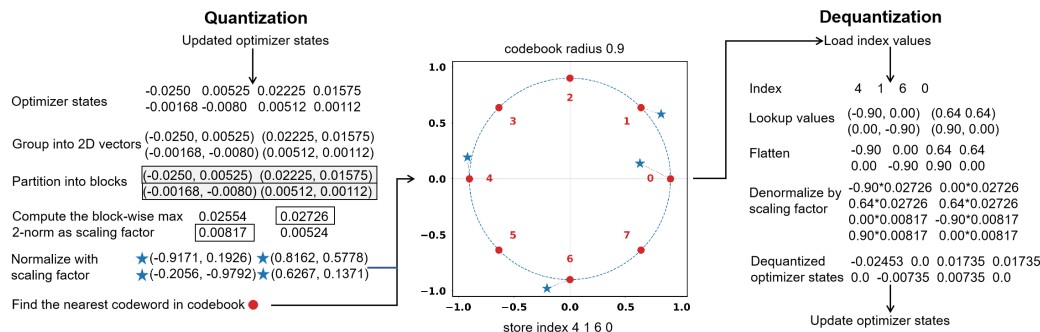

Figure 2: Schematic of our 2D polar quantization for optimizer states, implemented in a 1.5-bit update cycle (a codeword in the 8-entry codebook jointly encodes two parameters, namely 1.5 bits per parameter). After a high-precision (16/32-bit) update, the state tensor is treated as 2D vectors and partitioned into disjoint blocks. For each block, a scaling factor—the maximum 2-norm of its vectors—is computed. Each vector is normalized by this factor and mapped to the nearest point in a circular codebook, with only the codebook index stored. Dequantization reverses the process by retrieving the codebook vector and reapplying the scaling factor. Compared to 1D methods, this 2D approach reduces quantization distortion at the same bit-rate (Sayood, 2017) and supports non-integer bit representations.

**Quantized Adafactor.** For another popular optimizer, Adafactor, we can follow almost the same spirit to quantize it and obtain its quantized version. To this end, we only add two extra steps for dequantization and quantization. Compared with AdamW, Adafactor inherently reduces memory by factorizing the second-moment matrix. Accordingly, we only need to quantize its first moment while ingnoring the factorized second-moment accumulators, $R_t$ and $C_t$, since their size are very small and account for much less memory cost than first moment. See more discussion in Appendix B.

### 3.2 2D QUANTIZATION

The success of low-precision optimizers hinges on the ability of the quantization algorithm to minimize approximation error. Without proper control, this error can accumulate during training, leading to instability and poor convergence. Therefore, we focus on matrix-based quantization, as optimizer state tensors can be readily reshaped into matrices. We first identify the shortcomings of current techniques and then introduce a novel 2D quantization method that provides more robust error control, enabling the practical application of low-precision optimizers.

**Quantization.** Given a real matrix $\boldsymbol{X} = [\boldsymbol{x}_1, \boldsymbol{x}_2, \ldots, \boldsymbol{x}_{n/k}] \in \mathbb{R}^{k \times \frac{n}{k}}$ where $k$ divides $n$, we can partition the columns of $\boldsymbol{X}$ into multiple disjoint blocks, and compute the maximum 2-norm of the column vectors within each block. Usually, the size of each block ($k$ times the number of vectors in any block) should be as close as possible to a value called block size. Let $\mathcal{M}(\boldsymbol{X})$ be a vector whose $i$-th element $\mathcal{M}(\boldsymbol{X})_i$ is the maximum 2-norm of the column vectors in the block containing $\boldsymbol{x}_i$. We can define the normalization operator $\mathcal{N} : \mathbb{R}^{k \times \frac{n}{k}} \to \mathbb{R}^{k \times \frac{n}{k}}$ as $\mathcal{N}(\boldsymbol{X})_i = \boldsymbol{x}_i / \mathcal{M}(\boldsymbol{X})_i$, where $\mathcal{N}(\boldsymbol{X})_i$ is the $i$-th column of matrix $\mathcal{N}(\boldsymbol{X})$.

We define an injective quantization mapping $\mathcal{R} : \mathbb{T}_{kb} \to \mathbb{R}^k$. The image of this mapping, $\mathcal{R}(\mathbb{T}_{kb})$, constitutes the **codebook**, and each vector within this set is a **codeword**. The indexing map $\mathcal{I} : \mathbb{R}^k \to \mathbb{T}_{kb}$ then finds the nearest codeword for any given input vector and returns its corresponding index, where $\mathbb{T}_{kb} = \{0, 1, \ldots, 2^{kb} - 1\}$. The implementation of $\mathcal{R}$ can be seen in Sec. 4.1, and the implementation of $\mathcal{I}$ is in Sec. 4.2. After normalizing $\boldsymbol{X}$ with $\mathcal{N}$, we can use mapping $\mathcal{I}$ to quantize each column of $\mathcal{N}(\boldsymbol{X})$ to a $kb$-bit number ($b$ bits per element). Let $\mathcal{I}(\mathcal{N}(\boldsymbol{X}))$ be a vector whose $i$-th element $\mathcal{I}(\mathcal{N}(\boldsymbol{X}))_i = \mathcal{I}(\mathcal{N}(\boldsymbol{X})_i)$. Now the $k$-dimensional $b$-bit quantizer $\mathcal{Q}$ for quantizing $\boldsymbol{X}$ is given by

$$\mathcal{Q} = (\mathcal{I} \circ \mathcal{N}, \mathcal{M}) : \mathbb{R}^{k \times \frac{n}{k}} \to \mathbb{T}_{kb}^{\frac{n}{k}} \times \mathbb{R}^{\frac{n}{k}}. \tag{5}$$

**Dequantization.** Given a $k$-dimensional $b$-bit quantizer $\mathcal{Q} = (\mathcal{I} \circ \mathcal{N}, \mathcal{M})$ for quantizing matrix $\boldsymbol{X} \in \mathbb{R}^{k \times \frac{n}{k}}$, the corresponding dequantizer $\mathcal{D}$ is a mapping defined as

$$\mathcal{D}(\mathcal{Q}(\boldsymbol{X})) = \mathcal{D}(\mathcal{I} \circ \mathcal{N}(\boldsymbol{X}), \mathcal{M}(\boldsymbol{X})) = \mathcal{R}(\mathcal{I} \circ \mathcal{N}(\boldsymbol{X})) \odot \mathcal{M}(\boldsymbol{X}) : \mathbb{T}_{kb}^{\frac{n}{k}} \times \mathbb{R}^{\frac{n}{k}} \to \mathbb{R}^{k \times \frac{n}{k}}, \tag{6}$$

Table 1: Comparison of AdamW momentum quantization errors for the `q_proj` weight in the 8th layer of a LLaMA-130M model. For each column, the best result is shown in **bold** and the second-best is underlined.

| Method | First momentum | | | Second momentum | | |
|---|---|---|---|---|---|---|
| | NRE-1 $\downarrow$ | NRE-2 $\downarrow$ | AE(°)$\downarrow$ | NRE-1 $\downarrow$ | NRE-2 $\downarrow$ | AE(°)$\downarrow$ |
| Linear-2 2-bit | 0.513 | 0.947 | 26.747 | 0.800 | 0.007 | 32.965 |
| Dynamic 2-bit | 0.435 | 0.579 | 24.632 | 0.782 | 0.007 | 32.860 |
| Ours 2-bit | **0.394** | **0.421** | **22.631** | **0.295** | **0.002** | **16.665** |
| Ours 1.5-bit | 0.528 | 0.807 | 31.788 | 0.379 | 0.003 | 21.875 |

where $\mathcal{R}(\mathcal{I} \circ \mathcal{N}(\boldsymbol{X}))_i = \mathcal{R}(\mathcal{I}(\mathcal{N}(\boldsymbol{X}))_i)$ is the $i$-th column of matrix $\mathcal{R}(\mathcal{I} \circ \mathcal{N}(\boldsymbol{X}))$.

Our generalized framework unifies different quantization approaches through the vector dimension parameter $k$. Standard 1D quantization (Dettmers et al., 2022; Li et al., 2023) corresponds to the special case where $k = 1$. In this setting, each "vector" $\boldsymbol{x}_i$ is simply a scalar. Consequently, our general normalization based on the maximum 2-norm naturally simplifies to the conventional method of dividing by the block's maximum absolute value, as for any scalar $s$, $\|s\|_2 \equiv |s|$. While effective at higher precisions, this scalar approach introduces significant error in ultra-low-bit scenarios (e.g., below 4 bits), where a 1D codebook's representational capacity is insufficient.

To overcome this limitation, we explore the case of $k = 2$ by leveraging joint quantization. This directly applies our generalized procedure: parameters are grouped into 2D vectors, normalized by the block-wise maximum 2-norm, and mapped to a 2D codebook. This complete procedure for our 2D ($k = 2$) quantizer, visualized in Fig. 2, provides a fundamentally richer representation than its counterpart of $k = 1$.

The theoretical basis for this advantage is well-established. For instance, Bucklew & Gallagher (1979) showed that for 2D data, using a 2D polar coordinate system for quantization results in a lower mean square error (MSE) than a standard 1D Cartesian approach (Max, 1960). This principle was later generalized by Swaszek & Thomas (1983), who designed methods for even higher dimensions ($k > 2$) to minimize error for spherically symmetric data distributions.

To provide empirical proof, we first introduce the metrics used to evaluate static quantization error. Given $\boldsymbol{X} \in \mathbb{R}^{k \times \frac{n}{k}}$, to measure the difference between $\boldsymbol{X}$ and $\boldsymbol{Y} = \mathcal{D}(\mathcal{Q}(\boldsymbol{X}))$, we use the reshaping function $h : \mathbb{R}^{k \times \frac{n}{k}} \to \mathbb{R}^{m \times \frac{n}{m}}$ to reorder elements in $\boldsymbol{X}$ and $\boldsymbol{Y}$. $h$ first vectorizes the input matrix via column-wise concatenation, then partitions the resulting vector into contiguous segments of length $m$, and finally reshapes these segments into the matrix in $\mathbb{R}^{m \times \frac{n}{m}}$. We define the $m$-dimensional normwise relative error ($m$-NRE) and the angle error (AE) at $\boldsymbol{X}$ as

$$m\text{-NRE} = \text{mean}_i \left( \frac{\|h(\boldsymbol{X})_i - h(\boldsymbol{Y})_i\|_2}{\|h(\boldsymbol{X})_i\|_2 + \varepsilon} \right), \quad \text{AE} = \arccos \left( \frac{\langle \boldsymbol{X}, \boldsymbol{Y} \rangle}{(\|\boldsymbol{X}\|_F \|\boldsymbol{Y}\|_F)} \right),$$

where $\text{mean}_i(a_i)$ is the average value of all possible $a_i$ ($i = 1, \ldots m$), and $\varepsilon$ is a small positive number.

We consider three common 1D quantization mappings: Linear power quantization (Li et al., 2023; Wang et al., 2024), Dynamic quantization (Dettmers, 2016; Dettmers et al., 2022), and NormalFloat (NF) based on Quantile quantization (Dettmers et al., 2023). Since NF is mainly tailored for weight quantization, our analysis focuses on Linear and Dynamic mappings. See their specifications and visualizations in Appendix C. As shown in Tab. 1, our 2D approach consistently achieves the lowest relative error and the highest cosine similarity. This advantage is further confirmed by the distribution maps in Appendix D, where our method better preserves the original data structure.

## 4 EFFICIENT DESIGN AND THEORETICAL ANALYSIS

### 4.1 IMPLEMENTATION OF THE QUANTIZATION MAPPINGS

We present the implementation of our 2D 1.5-bit and 2.0-bit quantizer for signed and unsigned tensors. Further, we also give the implementation of our low-bit AdamW/Adafactor used for neural

network training. Let $\mathcal{Q} = (\mathcal{I} \circ \mathcal{N}, \mathcal{M})$ be a $k$D quantizer and $\mathcal{D}$ be its corresponding dequantizer as described in Sec. 3.2.

**2D quantization mappings for signed inputs.** Our 2D quantization mappings, illustrated in Fig. 2, are designed for superior representation efficiency. A straightforward method for creating a 2D codebook is to arrange points in a simple Cartesian grid (Fig. 1(a)). However, this approach is inherently suboptimal, as it treats each dimension independently and fails to leverage the primary benefit of joint quantization: modeling the correlation between values.

We therefore construct our codebook using a polar representation, placing points on concentric circles. All configurations share a common set of eight angles, $\Theta = \{j\pi/4 \mid j = 0, \ldots, 7\}$. The number of bits is determined by the radii $R$: a single radius ($R = \{0.40\}$) is used for 1.5-bit, while two radii ($R = \{0.14, 0.53\}$) are used for 2.0-bit. The resulting codebook is shown in Fig. 1(c), and the following lemma formalizes the advantages of this spherically symmetric sampling.

**Lemma 1.** *Let $\boldsymbol{x} \in \mathbb{R}^2, Y \subseteq \mathbb{R}^2$ and $s > 0$. If $\forall \boldsymbol{y} \in Y$, $\|\boldsymbol{x}\|_2 = s\|\boldsymbol{y}\|_2 > 0$ and the angle between $\boldsymbol{x}$ and $\boldsymbol{y}$ does not exceed $\phi \leq \frac{\pi}{2}$, then we have*

$$\|\boldsymbol{x} - \boldsymbol{y}\|_2 \leq \frac{2\sin(\phi/2) + |s - 1|}{s}\|\boldsymbol{x}\|_2.$$

The above lemma indicates that the relative quantization error can be controlled well by spherically symmetric sampling when $s \approx 1$. In our settings, $\phi = \frac{\pi}{8}$ and $\|\boldsymbol{x} - \boldsymbol{y}\|_2 \leq \frac{\pi}{8}\|\boldsymbol{x}\|_2$ if $s = 1$. The proof of Lemma 1 can be found in Appendix F.

**2D quantization mappings for unsigned inputs.** For unsigned inputs, such as the second moments in AdamW, a different strategy is required. Since these values are always non-negative, we enhance efficiency by concentrating all codebook points in the first quadrant.

A critical design choice is to avoid placing points on the origin or the axes. Mapping to zero can cause numerical instability during training, especially when a quantized value appears in a denominator (Li et al., 2023). We prevent this by ensuring all angles are sampled strictly between 0 and $\pi/2$.

Furthermore, unlike the uniform circular design for signed inputs, this mapping is non-uniform, meaning the number of available angles depends on the radius. For instance, our **1.5-bit** mapping uses three radii, $R = \{0.20, 0.42, 1.00\}$, with 2, 3, and 3 angles assigned to them respectively, while the **2.0-bit** mapping uses four radii, $R = \{0.20, 0.33, 0.53, 1.00\}$, with a corresponding 2, 4, 5, and 5 angles. This specialized structure can more robustly and effectively cover the positive data space.

### 4.2 IMPLEMENTATION OF THE QUANTIZERS

Given a quantization mapping $\mathcal{R} : \mathbb{T}_{kb} \to \mathbb{R}^k$, the simplest implementation of $\mathcal{I} : \mathbb{R}^k \to \mathbb{T}_{kb}$ is searching for the nearest neighbor of the input in $\mathcal{R}(\mathbb{T}_{kb})$, that is

$$\mathcal{I}(\boldsymbol{x}) = \mathrm{argmin}_{j \in \mathbb{T}_{kb}} \|\boldsymbol{x} - \mathcal{R}(j)\|_{p_2}, \tag{7}$$

where $p_2 = 1$ in our experiments. The time cost of Eqn. (7) is affordable when $b \leq 2$. If $b$ is large and $k > 1$, Eqn. (7) becomes time-consuming. Tian et al. (2025) propose a fast 2-dimensional quantization method with linear time complexity independent of quantization precision. Here we further develop it with more rigorous proofs and flexible mappings.

Consider mapping $f : \mathbb{R} \to \mathbb{C}$ defined as $f(t) = e^{i\theta} + e^{it\theta}$, where $\theta$ is an irrational number and $i = \sqrt{-1}$. We first introduce the concept of dense set, Dirichlet's approximation theorem and Kronecker's approximation theorem.

**Definition 1.** *Let $A \subseteq B \subseteq \mathbb{R}^n$, we say $A$ is dense in $B$ if the closure of $A$ is $B$. Equivalently $A$ is dense in $B$ if for any $\boldsymbol{x} \in B$, every neighborhood $U$ of $\boldsymbol{x}$ intersects $A$, that is, $U \cap A \neq \emptyset$.*

**Theorem 1** (Dirichlet). *Let $\alpha$ be a real number, and $k$ be a positive integer. Then there exist $p, q \in \mathbb{Z}$, such that $1 \leq q \leq k$ and $|\alpha - \frac{p}{q}| \leq \frac{1}{qk}$.*

**Corollary 1.** *Let $\alpha$ be a real number, then $\alpha$ is an irrational number if and only if $\forall \varepsilon > 0$, there exist $x, y \in \mathbb{Z}$ such that $0 < |\alpha x - y| < \varepsilon$.*

**Theorem 2** (Kronecker). *Let $\alpha$ be a real number, and $k$ be a positive integer. Then there exist $p, q \in \mathbb{Z}$, such that $1 \leq q \leq k$ and $|\alpha - \frac{p}{q}| \leq \frac{1}{qk}$.*

**Corollary 2.** *Let $\alpha$ be a real number, and $\theta$ be an irrational number, then $\forall \varepsilon > 0$, there exist $n, k \in \mathbb{Z}$ such that $|n\theta - k - \alpha| < \varepsilon$.*

The proofs of Theorem 1 and Theorem 2 can be found in Appendix F. $\forall \varepsilon > 0$, we can obtain an algorithm with constant time complexity to compute $\tilde{t} \in \mathbb{R}$ such that $|f(\tilde{t}) - z| < \varepsilon$, where $|z| \leq 2$. To see this, let $z = x_0 + iy_0 = r\cos\varphi + ir\sin\varphi$, where $x_0, y_0 \in \mathbb{R}$ and $r = |z| \in [0, 2]$. According to Corollary 2, $\forall \mu > 0$ there exist $n_1, n_2 \in \mathbb{Z}$ such that

$$\left| \frac{1+\theta}{1-\theta} n_1 - n_2 - \frac{\varphi - \frac{1+\theta}{1-\theta}\arccos\frac{r}{2}}{2\pi} \right| < \mu \Rightarrow \left| \frac{1+\theta}{1-\theta}\left(2\arccos\frac{r}{2} + 4n_1\pi\right) - (2\varphi + 4n_2\pi) \right| < 4\pi\mu.$$

Note that the proofs of Theorem 1 and Theorem 2 are constructive, thus it is easy to see that $n_1, n_2$ can be obtained within constant time complexity. Let $f(t) = x(t) + iy(t)$, where $x(t), y(t)$ are real functions. Define $\tilde{t} = \frac{2\arccos\frac{r}{2} + 4n_1\pi}{1-\theta}$, we can prove that

$$|x_0 - x(\tilde{t})| < 2r\pi\mu \leq 4\pi\mu, \quad |y_0 - y(\tilde{t})| < 2r\pi\mu \leq 4\pi\mu.$$

Specifically, if $\theta = 1 - 4\pi$, we have $\tilde{t} = n_1 + \frac{\arccos\frac{r}{2}}{2\pi}$, and $\frac{\arccos\frac{r}{2}}{2\pi} \in [0, 0.25]$. In this case, the integer part of $\tilde{t}$ reflects the angle information of $f(\tilde{t})$, and the fractional part of $\tilde{t}$ reflects the magnitude information of $f(\tilde{t})$. The number of all possible values of $n_1$ does not exceed $1 + 1/\mu$. If the range of $r$ is discrete, a fast quantization method can be derived from the above discussion. In summary, we get the following theorem.

**Theorem 3.** *Suppose mapping $f : \mathbb{R} \to \mathbb{C}$ is defined as $f(t) = e^{i\theta} + e^{it\theta}$, where $\theta$ is an irrational number. Then $f(\mathbb{R})$ is dense in $\{z \in \mathbb{C} \mid |z| \leq 2\}$.*

## 5 EXPERIMENTS

Now we compare our ultra-low-bit optimizers against their 16/32-bit counterparts, a 4-bit baseline (Li et al., 2023), and its naive 2-bit extension, across LLM pretraining and fine-tuning, and vision training tasks on a single A800 GPU. To ensure fairness, we keep the weights, gradients, activations, and hyperparameters identical to the public baselines, replacing only the optimizer with our 1.5-bit or 2-bit variants. Medium-scale LLMs and vision models are pretrained with 16/32-bit optimizers as baselines, while LLaMA2-7B fine-tuning uses 8-bit Adafactor as the high-precision reference.

**Models and datasets.** We pretrain GPT-2 (124M) for 40k steps on OpenWebText (Gokaslan & Cohen, 2019) following the nanoGPT codebase, and LLaMA-2 (130M, 350M) for 80k steps on C4 (Raffel et al., 2020) following (Zhao et al., 2024). For fine-tuning, LLaMA2-7B and Qwen2.5-7B are trained on Alpaca (Taori et al., 2023) and evaluated with GLUE (Wang et al., 2019) using lm-evaluation-harness (Gao et al., 2024). For vision tasks, we use codebase in (Zhou et al., 2023) to train ViT-B/16 (Dosovitskiy et al., 2021) and ResNet-50 (He et al., 2016) on ImageNet-1K (Russakovsky et al., 2015) for 150 and 100 epochs, respectively. Further details are in Appendix G.

**Quantization setup.** Optimizer states $X$ are partitioned into blocks of size 64 and quantized using our 2D $k$-bit quantizers (see Sec. 4.1). For additional compression, $\mathcal{M}(X)$ is dynamically quantized to 8-bit with block size 256, following the double-quantization scheme of (Dettmers et al., 2023). For Adafactor, we adopt a learning rate scaling factor $\alpha = 2.0$. For AdamW, we set $\alpha = 2.0$ for 2-bit quantization and $\alpha = 2.5$ for 1.5-bit quantization (see Sec. 3.1). All other optimizer hyperparameters remain unchanged.

### 5.1 MAIN RESULTS

On LLMs, we report the performance, wall-clock time, and memory cost of optimizers in Tab. 2. Importantly, Tab. 2 shows that a naive extension of the 4.0-bit baseline's 1D quantization technique to 2 bits consistently leads to training collapse (*Crash*). In contrast, our proposed method not only ensures stable training but also demonstrates significant advantages in both efficiency and performance. Specifically, our method enables stable quantization down to 2.0 and even 1.5 bits, achieving up to a **4.6×** reduction in optimizer memory footprint compared to the 16-bit baseline while maintaining highly competitive wall-clock times. Fig. 3 also shows the validation perplexity

Table 2: Comparison of validation perplexity (VPPL), wall-clock time (WCT, hours), and GPU Memory usage of optimizer states (MC, MB). We train LLAMA-130M/350M on the C4 dataset, and GPT2-124M on OpenWebText. *Crash* indicates training failures (NaN loss or non-convergence).

| Optimizer | LLAMA-130M | | | LLAMA-350M | | | GPT2-124M | | |
|---|---|---|---|---|---|---|---|---|---|
| | VPPL | WCT | MC | VPPL | WCT | MC | VPPL | WCT | MC |
| 32-bit Adafactor | 20.393 | 22.98 | 516.87 | 17.708 | 59.57 | 1429.64 | 19.556 | 31.34 | 476.45 |
| 4.0-bit Adafactor | 20.497 | 22.99 | 228.42 | 17.481 | 59.70 | 411.77 | 19.658 | 31.30 | 200.23 |
| 2.0-bit Adafactor | *Crash* | - | - | *Crash* | - | - | *Crash* | - | - |
| 2.0-bit Adafactor(Ours) | 20.243 | 23.04 | 208.80 | 16.790 | 59.98 | 330.66 | 19.890 | 31.37 | 173.75 |
| 1.5-bit Adafactor(Ours) | 20.273 | 23.14 | 203.74 | 16.712 | 59.50 | 312.63 | 20.164 | 31.32 | 168.69 |
| 16-bit AdamW | 20.350 | 22.33 | 518.70 | 16.866 | 58.36 | 1420.72 | 19.645 | 31.30 | 951.90 |
| 4.0-bit AdamW | 20.680 | 22.49 | 269.30 | 17.123 | 59.04 | 582.98 | 19.842 | 31.29 | 391.46 |
| 2.0-bit AdamW | *Crash* | - | - | *Crash* | - | - | *Crash* | - | - |
| 2.0-bit AdamW(Ours) | 20.480 | 22.63 | 230.06 | 16.917 | 59.09 | 403.66 | 19.979 | 31.34 | 341.50 |
| 1.5-bit AdamW(Ours) | 20.904 | 22.59 | 219.94 | 17.157 | 59.14 | 367.62 | 20.665 | 31.39 | 331.38 |

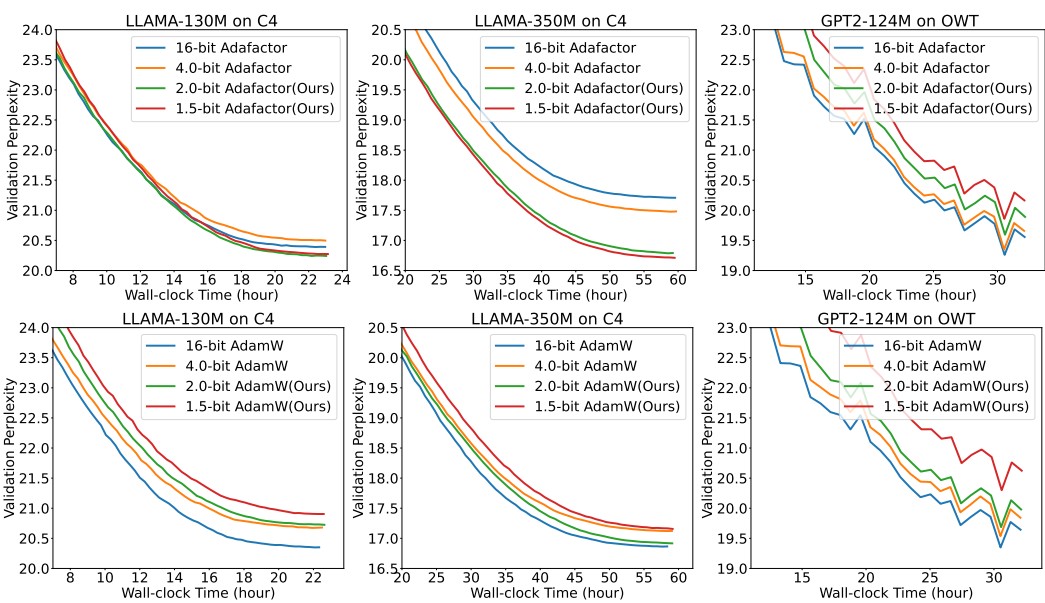

Figure 3: Validation perplexity curves on the C4 and OWT datasets.

curves of pretraining LLMs on the C4 and OWT datasets, and demonstrates of superiority of our method.

Notably, our method shows exceptional strength in certain settings; for instance, during the LLaMA-350M pretraining with AdamW, our 2-bit optimizer not only outperforms the standard 4-bit baseline but also closes **over 80%** of the performance gap to the full 16-bit version. Even more strikingly, for Adafactor pretraining on both LLaMA-130M and LLaMA-350M, the validation perplexity for our low-bit optimizer eventually surpasses the high-precision baseline in the later stages of training. We hypothesize this phenomenon is due to the inherent regularization effect of ultra-low-bit quantization, where the introduced noise may help the optimizer escape sharp local minima and settle into flatter, more generalizable minima.

We also conducted experiments on vision pretraining tasks using ViT-Base/16 and ResNet-50 models. The results, summarized in Tab. 3, reinforce the findings from our LLM experiments. Once again, a naive extension to 2-bit quantization consistently fails, while our method ensures stable training. Remarkably, our approach delivers substantial efficiency gains, particularly for ViT-Base/16, where it reduces the AdamW optimizer memory footprint from 654.4 MB to just 49.4 MB—a reduction of over **13×**—with only a minor ∼1% drop in Top-1 accuracy. Furthermore, in AdamW experiments

Table 3: Test accuracy (TA), wall-clock time (WCT, hours), and GPU Memory usage of optimizer states (MC, MB) on ImageNet-1K classification. *Crash* indicates training failures.

| Optimizer | ViT-Base/16 | | | ResNet-50 | | |
|---|---|---|---|---|---|---|
| | TA | WCT | MC | TA | WCT | MC |
| 32-bit SGDM | - | - | - | 75.41 | 29.94 | 94.21 |
| 32-bit Adafactor | 80.72 | 58.80 | 326.11 | 77.60 | 31.99 | 214.71 |
| 4.0-bit Adafactor | 80.57 | 59.19 | 59.09 | 76.85 | 32.37 | 135.66 |
| 2.0-bit Adafactor | *Crash* | - | - | *Crash* | - | - |
| 2.0-bit Adafactor(Ours) | 79.53 | 59.35 | 26.25 | 76.72 | 32.39 | 125.89 |
| 32-bit AdamW | 80.72 | 56.47 | 654.38 | 77.68 | 30.06 | 192.87 |
| 4.0-bit AdamW | 79.28 | 56.85 | 103.39 | 75.65 | 31.24 | 24.91 |
| 2.0-bit AdamW | *Crash* | - | - | *Crash* | - | - |
| 2.0-bit AdamW(Ours) | 79.68 | 57.32 | 49.39 | 76.30 | 31.21 | 10.93 |

Table 4: Performance of fine-tuned LLMs on the GLUE benchmark during fine-tuning on the Alpaca dataset. AVG = average GLUE score. TMC = total memory cost (TMC).

| Model | Optimizer | SST-2 | RTE | COLA | MNLI | MRPC | AVG | TMC (MB) |
|---|---|---|---|---|---|---|---|---|
| LLaMA2-7B | Original | 83.0 | 70.8 | 29.2 | 36.5 | 66.9 | 57.3 | - |
| | 8.0-bit Adafactor | 91.5 | 74.7 | 35.1 | 53.9 | 65.0 | 64.0 | 54 220 |
| | 1.5-bit Adafactor | 91.3 | 74.0 | 35.7 | 50.0 | 68.1 | 63.8 | 49 188 |
| Qwen2.5-7B | Original | 94.8 | 83.0 | 46.0 | 77.3 | 75.3 | 75.3 | - |
| | 8.0-bit Adafactor | 94.6 | 80.1 | 46.6 | 70.4 | 76.5 | 73.6 | 62 969 |
| | 1.5-bit Adafactor | 95.4 | 80.9 | 47.6 | 71.4 | 75.7 | 74.5 | 57 929 |

on both models, our 2-bit optimizer again outperforms the 4-bit baseline, confirming its superior balance of compression and performance. Wall-clock times remained competitive across all settings, indicating minimal computational overhead.

In Tab. 4, we can see the performance of fine-tuned LLMs on the GLUE benchmark. These results further verify the effectiveness and scalability of our proposed low-bit Adafactor.

## 5.2 ABLATION STUDIES

We investigate the impact of different values of hyperparameters scale factor $\alpha$ on performance. The results shown in Tab. 5 demonstrate the rationality of our chosen hyperparameters.

Table 5: Validation perplexity (VPPL) of training LLAMA-130M on the C4 dataset and GPT-2 (124M) on the OpenWebText dataset.

| Model | Optimizer | VPPL |
|---|---|---|
| LLAMA -130M | 1.5-bit Adafactor ($\alpha = 1.0$) | 20.543 |
| | 1.5-bit Adafactor ($\alpha = 2.0$) | 20.273 |
| | 1.5-bit AdamW ($\alpha = 2.0$) | 21.072 |
| | 1.5-bit AdamW ($\alpha = 2.5$) | 20.904 |
| | 1.5-bit AdamW ($\alpha = 3.0$) | 21.036 |
| GPT2 -124M | 1.5-bit AdamW ($\alpha = 2.0$) | 20.648 |
| | 1.5-bit AdamW ($\alpha = 2.5$) | 20.665 |
| | 1.5-bit AdamW ($\alpha = 3.0$) | 20.624 |

## 6 CONCLUSIONS, LIMITATIONS, AND BROADER IMPACT

We propose 2.0-bit and 1.5-bit variants of AdamW and Adafactor, based on a novel 2D quantization framework and its associated design principles. Experimental results on a diverse set of benchmarks demonstrate that our low-bit optimizers maintain comparable performance to their high-precision counterparts while substantially reducing memory consumption.

**Limitations & Broader impact.** Due to limitations in computing resources, we did not pretrain our ultra-low-bit optimizers on large-scale models with more than 1B parameters. For impact, our research opens up the field of using multi-dimensional quantization to realize ultra-low-bit optimizers for LLM training. It benefits AI researchers with limited GPU memory resources.

## ETHICS STATEMENT

All datasets used in this work are publicly available and widely used in the research community (e.g., C4, OpenWebText). No private, proprietary, or personally identifiable data were collected or used in our experiments. We have adhered to the licensing terms of each dataset and ensured that the data were processed in compliance with ethical standards. Additionally, we are committed to ensuring fairness and impartiality in the research process, avoiding any form of bias.

## REPRODUCIBILITY STATEMENT

To facilitate reproducibility, we provide detailed experimental settings in Sec. 5 and Appendix G. We also release an anonymous code repository at https://anonymous.4open.science/r/ultra-low-bit-optimizers.

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

## A    DECLARATION OF LLM USAGE

During the preparation of this work, we used Gemini 2.5 Pro Think and GPT-5 to polish the English expression and check for spelling errors in our manuscript. No parts of the core research ideas, methods, results, or conclusions were generated by LLMs. All experimental code and data analysis were conducted and verified by the authors.

## B    LOW-BIT ADAMW AND ADAFACTOR

Our quantization strategy is applied selectively to specific parameter groups within the model. We define $S_Q$ as the set of indices for parameter groups designed for quantization. In our experiments, $S_Q$ primarily contains the core weight tensors of linear layers (which are typically 2D-shaped), as these are the most memory-intensive.

Conversely, several parameter groups are intentionally kept in full precision and are thus excluded from $S_Q$. For Large Language Models (LLMs), these non-quantized parts notably include the embedding layer parameters, the bias vectors within linear layers, and all parameters within normalization layers (e.g., LayerNorm, RMSNorm).

For all parameter groups $\boldsymbol{\theta}_i$, we introduce a layer-wise scaling factor $\alpha_i$, which is applied to the normalized momentum term before the main parameter update. This factor is formally defined as:

$$\alpha_i = \begin{cases} \alpha_{\text{scale}} & \text{if } i \in S_Q \\ 1.0 & \text{if } i \notin S_Q \end{cases} \tag{8}$$

In contrast, previous approaches (Dettmers et al., 2022; Li et al., 2023) that do not use such a mechanism are equivalent to setting $\alpha_i = 1.0$ for all layers. The hyperparameter $\alpha_{\text{scale}}$ is set based on the quantization level (e.g., 2.0 for 2-bit). The modified optimization procedures are detailed in Algorithms 2 and 3.

---

**Algorithm 2** AdamW with quantized states

**Input:** Steps $T$, learning rate $\eta$, moment decays $\beta_1, \beta_2$, weight decay $\lambda$, epsilon $\epsilon$
1: Let $S_Q$ be the set of quantized layer indices
2: **for** each parameter group $i$ **do**
3:    $\boldsymbol{\theta}_{0,i}, \boldsymbol{m}_{0,i}^q \leftarrow 0, \boldsymbol{v}_{0,i}^q \leftarrow 0$
4: **for** $t = 1, \ldots, T$ **do**
5:    **for** each parameter group $i$ **do**
6:       $\boldsymbol{g}_{t,i} \leftarrow \nabla_{\boldsymbol{\theta}} f(\boldsymbol{\theta}_{t-1,i})$
7:       **if** $i \in S_Q$ **then**
8:          $\boldsymbol{m}_{t-1,i}, \boldsymbol{v}_{t-1,i} \leftarrow$ Dequantize$(\boldsymbol{m}_{t-1,i}^q, \boldsymbol{v}_{t-1,i}^q)$
9:       **else**
10:          $\boldsymbol{m}_{t-1,i}, \boldsymbol{v}_{t-1,i} \leftarrow \boldsymbol{m}_{t-1,i}^q, \boldsymbol{v}_{t-1,i}^q$ {States are not quantized}
11:       $\boldsymbol{m}_{t,i} \leftarrow \beta_1 \boldsymbol{m}_{t-1,i} + (1 - \beta_1)\boldsymbol{g}_{t,i}$
12:       $\boldsymbol{v}_{t,i} \leftarrow \beta_2 \boldsymbol{v}_{t-1,i} + (1 - \beta_2)\boldsymbol{g}_{t,i}^2$
13:       $\boldsymbol{m}_{t,i}^c \leftarrow \boldsymbol{m}_{t,i}/(1 - \beta_1^t)$
14:       $\boldsymbol{v}_{t,i}^c \leftarrow \boldsymbol{v}_{t,i}/(1 - \beta_2^t)$
15:       Define $\alpha_i$ as in Eqn. (8)
16:       $\boldsymbol{u}_{t,i} \leftarrow \alpha_i \frac{\boldsymbol{m}_{t,i}^c}{\sqrt{\boldsymbol{v}_{t,i}^c} + \epsilon} + \lambda \boldsymbol{\theta}_{t-1,i}$
17:       $\boldsymbol{\theta}_{t,i} \leftarrow \boldsymbol{\theta}_{t-1,i} - \eta_t \cdot \boldsymbol{u}_{t,i}$
18:       **if** $i \in S_Q$ **then**
19:          $\boldsymbol{m}_{t,i}^q, \boldsymbol{v}_{t,i}^q \leftarrow$ Quantize$(\boldsymbol{m}_{t,i}, \boldsymbol{v}_{t,i})$
20:       **else**
21:          $\boldsymbol{m}_{t,i}^q, \boldsymbol{v}_{t,i}^q \leftarrow \boldsymbol{m}_{t,i}, \boldsymbol{v}_{t,i}$

---

**Algorithm 3** Adafactor with quantized states

**Input:** Steps $T$, learning rate $\eta$, decays $\beta_1, \beta_2$, epsilons $\epsilon_1, \epsilon_2$
1: Let $S_Q$ be the set of quantized layer indices
2: **for** each parameter group $i$ **do**
3:    $\boldsymbol{\theta}_{0,i}, \boldsymbol{m}_{0,i}^q \leftarrow 0, \boldsymbol{R}_{0,i} \leftarrow 0, \boldsymbol{C}_{0,i} \leftarrow 0$
4: **for** $t = 1, \ldots, T$ **do**
5:    **for** each parameter group $i$ **do**
6:       $\boldsymbol{g}_{t,i} \leftarrow \nabla_{\boldsymbol{\theta}} f(\boldsymbol{\theta}_{t-1,i})$
7:       **if** $i \in S_Q$ **then**
8:          $\boldsymbol{m}_{t-1,i} \leftarrow$ Dequantize$(\boldsymbol{m}_{t-1,i}^q)$
9:       **else**
10:          $\boldsymbol{m}_{t-1,i} \leftarrow \boldsymbol{m}_{t-1,i}^q$
11:       $\boldsymbol{m}_{t,i} \leftarrow \beta_1 \boldsymbol{m}_{t-1,i} + (1 - \beta_1)\boldsymbol{g}_{t,i}$
12:       $\boldsymbol{R}_{t,i} \leftarrow \beta_2 \boldsymbol{R}_{t-1,i} + (1 - \beta_2)\text{E}_{\text{row}}[\boldsymbol{g}_{t,i}^2]$
13:       $\boldsymbol{C}_{t,i} \leftarrow \beta_2 \boldsymbol{C}_{t-1,i} + (1 - \beta_2)\text{E}_{\text{col}}[\boldsymbol{g}_{t,i}^2]$
14:       $\boldsymbol{m}_{t,i}^c \leftarrow \boldsymbol{m}_{t,i}/(1 - \beta_1^t)$
15:       $\boldsymbol{V}_{t,i} \leftarrow (\boldsymbol{R}_{t,i} \otimes \boldsymbol{C}_{t,i})/\text{mean}(\boldsymbol{R}_{t,i})$
16:       $\eta_t \leftarrow \max(\epsilon_2, \text{RMS}(\boldsymbol{\theta}_{t-1,i}))^{-1}$
17:       Define $\alpha_i$ as in Eqn. (8)
18:       $\boldsymbol{u}_{t,i} \leftarrow \alpha_i \frac{\boldsymbol{m}_{t,i}^c}{\sqrt{\boldsymbol{V}_{t,i}} + \epsilon_1}$
19:       $\boldsymbol{\theta}_{t,i} \leftarrow \boldsymbol{\theta}_{t-1,i} - \eta_t \cdot \boldsymbol{u}_{t,i}$
20:       **if** $i \in S_Q$ **then**
21:          $\boldsymbol{m}_{t,i}^q \leftarrow$ Quantize$(\boldsymbol{m}_{t,i})$
22:       **else**
23:          $\boldsymbol{m}_{t,i}^q \leftarrow \boldsymbol{m}_{t,i}$

---

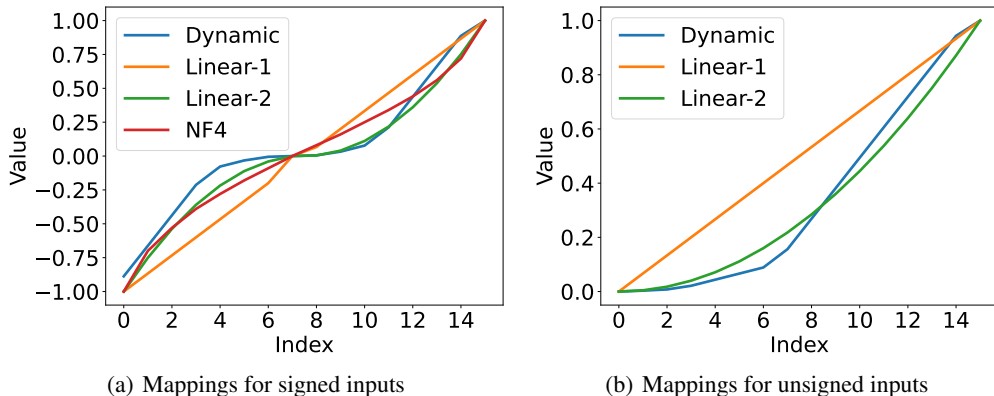

(a) Mappings for signed inputs        (b) Mappings for unsigned inputs

Figure 4: Visualization of different 4-bit quantization mappings for signed and unsigned inputs.

## C    1D QUANTIZATION MAPPINGS

We present the constructions of different quantization mappings in 1D $b$-bit quantizers ($\mathcal{R}$ in $\mathcal{Q}$). See Fig. 4 for their illustration. Note that $\mathbb{T}_b = \{0, 1, \ldots, 2^b - 1\}$.

**Linear power quantization.** Linear power (Linear-$p$) quantization for signed inputs is defined as

$$\mathcal{R}(j) = \begin{cases} -\left(-1 + 2j/(2^b - 1)\right)^p, & j < 2^{b-1} - 1; \\ 0, & j = 2^{b-1} - 1; \\ \left(-1 + 2j/(2^b - 1)\right)^p, & j > 2^{b-1} - 1, \end{cases}$$

where $j \in \mathbb{T}_b$ and $p > 0$. For unsigned inputs, it is defined as

$$\mathcal{R}(j) = \left(j/(2^b - 1)\right)^p, \quad j \in \mathbb{T}_b, p > 0.$$

**Dynamic quantization.** Dynamic quantization $\mathcal{R}$ for $b$-bit quantization maps $\mathbb{T}_b$ onto $\{0, 1\} \cup G$. For signed inputs, $G$ is a set of numbers with the following properties: the number in $G$ looks like $\pm q_k \times 10^{-E}$, where

$$\begin{cases} b = 2 + E + F, & E, F \in \mathbb{N}; \\ q_k = (p_k + p_{k+1})/2, & k \in \{0, \ldots, 2^F - 1\}; \\ p_j = 0.9j/2^F + 0.1, & j \in \{0, \ldots, 2^F\}. \end{cases}$$

For unsigned inputs, $G$ is a set of numbers with: the number in $G$ looks like $q_k \times 10^{-E}$, where

$$\begin{cases} b = 2 + E + F, & E, F \in \mathbb{N}; \\ q_k = (p_k + p_{k+1})/2, & k \in \{0, \ldots, 2^{F+1} - 1\}; \\ p_j = 0.9j/2^F + 0.1, & j \in \{0, \ldots, 2^{F+1}\}. \end{cases}$$

**NormalFloat.** $b$-bit NormalFloat $\mathcal{R}$ is built on Quantile quantization. range($\mathcal{R}$) is constructed as follows: evenly sampling $2^b$ quantiles of the standard normal distribution at first, and then normalizing them into $[-1, 1]$. For 4-bit NormalFloat (NF4) handling signed inputs, range($\mathcal{R}$) is about {-1.00, -0.70, -0.53, -0.39, -0.28, -0.18, -0.09, 0.00, 0.08, 0.16, 0.25, 0.34, 0.44, 0.56, 0.72, 1.00}.

## D    THE DISTRIBUTION OF OPTIMIZER STATES

To motivate our approach and highlight the challenges of low-bit optimizer state quantization, we first analyze the empirical distributions of AdamW's momentum states. As illustrated in Fig. 5, we plot the distributions for both the first momentum (Fig. 5(a)) and the second momentum (Fig. 5(b)), captured from the `q_proj` weight of the 8th layer in a LLaMA-130M model during its training. A

key observation is the stark difference between the two distributions: the first momentum exhibits a quasi-Gaussian distribution, whereas the second momentum is non-negative and highly skewed, with a high concentration of values near zero.

This pronounced asymmetry in the second momentum suggests that data-oblivious quantization schemes, such as uniform quantization, are highly inefficient. Based on this insight, We compare our proposed 2D quantization method against a baseline 1D quantization scheme using Kernel Density Estimation (KDE). Fig. 6 clearly demonstrates that our method more faithfully reproduces the original full-precision distribution for both states.

Crucially, this superiority is particularly striking for the highly-skewed second momentum and persists even under an aggressive 1.5-bit quantization, a regime where the 1D method fails to capture the essential data structure.

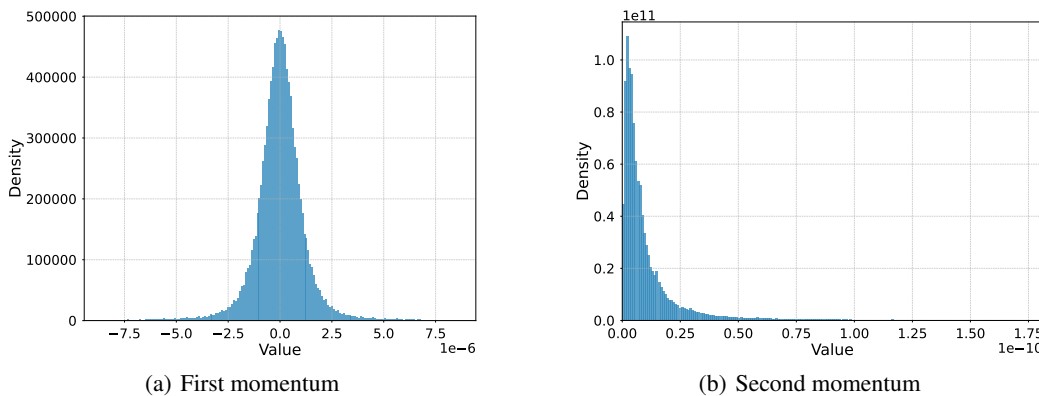

(a) First momentum

(b) Second momentum

Figure 5: Empirical distributions of AdamW states for the `q_proj` weight in the 8th layer of a LLaMA-130M model.

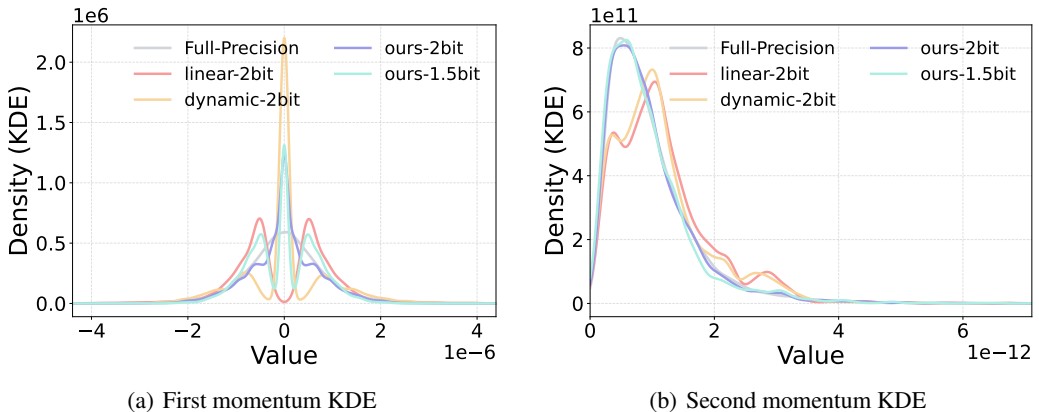

(a) First momentum KDE

(b) Second momentum KDE

Figure 6: A comparison of quantization methods on the first-order and second-order momentum states for the `q_proj` weight in the 8th layer of a LLaMA-130M model. The full-precision data serves as the ground truth. The Kernel Density Estimation (KDE) plots visualize the data's probability distribution with a smooth curve, allowing a qualitative assessment of how well each method preserves the original shape. As shown, data-aware approaches, such as our proposed 2D quantization method, more faithfully reproduce the original distribution compared to the data-oblivious 1D quantization scheme. This advantage is particularly pronounced for the highly-skewed second-order momentum, and it holds true even under an aggressive 1.5-bit quantization.

## E   MORE EXPERIMENTS RESULTS

Fig. 7 presents the training loss curves for fine-tuning on the Alpaca dataset. The convergence trajectory of our low-bit method closely tracks that of the full-precision baseline, indicating negligible performance degradation.

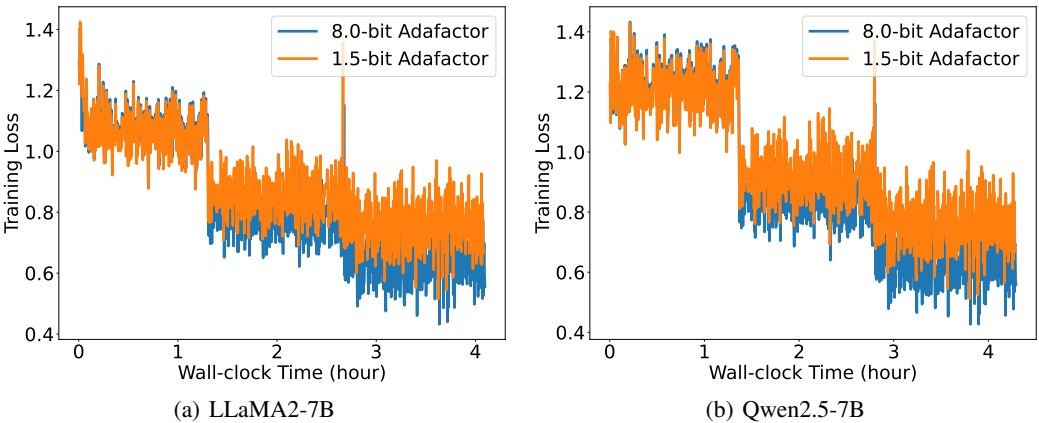

(a) LLaMA2-7B                    (b) Qwen2.5-7B

Figure 7:  Training loss curves of fine-tuning LLMs on the Alpaca dataset.

## F   PROOFS

**Lemma 1.** *Let $\boldsymbol{x} \in \mathbb{R}^2, Y \subseteq \mathbb{R}^2$ and $s > 0$. If $\forall \boldsymbol{y} \in Y$, $\|\boldsymbol{x}\|_2 = s\|\boldsymbol{y}\|_2 > 0$ and the angle between $\boldsymbol{x}$ and $\boldsymbol{y}$ does not exceed $\phi \leq \frac{\pi}{2}$, then we have*

$$\|\boldsymbol{x} - \boldsymbol{y}\|_2 \leq \frac{2\sin(\phi/2) + |s - 1|}{s}\|\boldsymbol{x}\|_2.$$

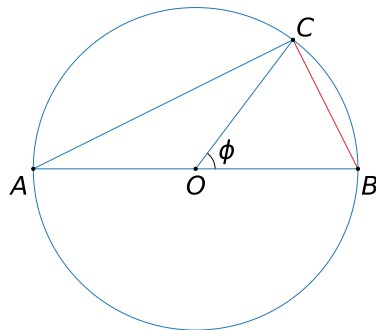

Figure 8:  Visualization of a circle centered at $O$ and its inscribed triangle $\triangle ABC$. The line segment $AB$ passes through point $O$, and $\phi = \angle COB$.

*Proof.* Consider Fig. 8. Without loss of generality, we assume that $\boldsymbol{y} = \overrightarrow{OB} \in Y$ and $\boldsymbol{x} = s\overrightarrow{OC}$. According to the properties of the inscribed triangle of a circle, we have

$$\|\overrightarrow{BC}\|_2 = 2\|\boldsymbol{y}\|_2 \sin(\phi/2).$$

The 2-norm satisfies the triangle inequality. Thus,

$$\|\boldsymbol{x} - \boldsymbol{y}\|_2 = \|\overrightarrow{OC} - \overrightarrow{OB} + (s-1)\overrightarrow{OC}\|_2$$
$$\leq \|\overrightarrow{BC}\|_2 + \|(s-1)\overrightarrow{OC}\|_2$$
$$= \frac{2\sin(\phi/2) + |s-1|}{s}\|\boldsymbol{x}\|_2.$$

The proof is completed. □

**Theorem 1** (Dirichlet). *Let $\alpha$ be a real number, and $k$ be a positive integer. Then there exist $p, q \in \mathbb{Z}$, such that $1 \leq q \leq k$ and $|\alpha - \frac{p}{q}| \leq \frac{1}{qk}$.*

*Proof.* Consider dividing the interval $[0,1]$ into $k$ sub-intervals $[0, \frac{1}{k}), \ldots, [\frac{k-2}{k}, \frac{k-1}{k}), [\frac{k-1}{k}, 1]$. Let real number $x = [x] + \{x\}$, where $[x]$ is the integer part of $x$, and $\{x\}$ is the fractional part of $x$ satisfying $0 \leq \{x\} < 1$. By the pigeonhole principle, among the $k + 1$ numbers $0, \{\alpha\}, \ldots, \{(k-1)\alpha\}, \{k\alpha\}$, there must be at least two numbers that lie in the same sub-interval among the aforementioned $k$ sub-intervals. Thus, there exist $0 \leq i < j \leq k$ such that $|\{j\alpha\} - \{i\alpha\}| \leq \frac{1}{k}$. Hence,

$$\left|\alpha - \frac{[j\alpha] - [i\alpha]}{j - i}\right| \leq \frac{1}{k(j-i)}.$$

The proof is completed. □

**Theorem 2** (Kronecker). *Let $\alpha$ be a real number, and $k$ be a positive integer. Then there exist $p, q \in \mathbb{Z}$, such that $1 \leq q \leq k$ and $|\alpha - \frac{p}{q}| \leq \frac{1}{qk}$.*

*Proof.* Since $n\theta = n[\theta] + n\{\theta\}$, we have $\{n\theta\} = \{n\{\theta\}\}$. Therefore, we can assume that $\theta \in (0,1)$. According to Corollary 1, for any $1 \geq \varepsilon > 0$, there exist $x, y \in \mathbb{Z}$ such that $0 < |\theta x - y| < \varepsilon$. Since $\theta > 0$, without loss of generality, we assume that $x, y$ are positive numbers.

1) If $\theta x > y$, we get $0 < \{\theta x\} < \varepsilon$. Thus, there exists positive integer $N$ such that $\frac{1}{N+1} < \{\theta x\} < \frac{1}{N} \leq \varepsilon$. For positive integer $k \leq N$, since $0 < k\{\theta x\} < 1$, we have $\{k\theta x\} = k\{\theta x\}$. This indicate that the $N$ numbers $\{\theta x\}, \ldots, \{N\theta x\}$ form an arithmetic sequence. Additionally, since

$$0 < 1 - N\{\theta x\} < 1 - \frac{N}{N+1} = \frac{1}{N+1} < \{\theta x\} < \varepsilon,$$

thus, those $N$ numbers divide the interval $[0,1]$ into $N+1$ sub-intervals, and each of them is no longer than $\varepsilon$.

2) If $\theta x < y$, we get $0 < 1 - \{\theta x\} < \varepsilon$. Thus, there exists positive integer $N$ such that $\frac{1}{N+1} < 1 - \{\theta x\} < \frac{1}{N} \leq \varepsilon$. For positive integer $k \leq N$, since $0 < k - k\{\theta x\} < 1$, we have $\{k\theta x\} = k\{\theta x\} - k + 1$. This indicate that the $N$ numbers $\{\theta x\}, \ldots, \{N\theta x\}$ form an arithmetic sequence. Additionally, since

$$0 < 1 - N + N\{\theta x\} < 1 - \frac{N}{N+1} = \frac{1}{N+1} < 1 - \{\theta x\} < \varepsilon,$$

thus, those $N$ numbers divide the interval $[0,1]$ into $N+1$ sub-intervals, and each of them is no longer than $\varepsilon$.

According to 1), 2) and the definition of density, the proof is completed. □

# G EXPERIMENTAL DETAILS

In our experiments, we use one A800 GPU under the PyTorch 2.2.0 + CUDA12.1 framework. To obtain the total peak memory consumption per GPU, we call "torch.cuda.max_memory_allocated". The total memory cost includes data, model parameters, activations, gradients, optimizer states and memory fragments. We calculate the memory cost of the optimizer states by taking the difference

between the memory usage of training with the target optimizer and the memory usage of training with a momentum-free optimizer.

For Adafactor, we set eps $= (10^{-30}, 10^{-3})$, clip_threshold $= 1.0$, decay_rate $= -0.8$ and $\beta_1 = 0.9$ by default. For AdamW, we set $\beta_1 = 0.9$ and $\beta_2 = 0.95$. For quantization settings, matrices with a size smaller than $4096$ will not be quantized.

**Settings on training LLAMA-2 on C4.** We run Adafactor/AdamW with 2000 warmup steps for training 130M LLAMA-2 and with 4000 warmup steps for training 350M LLAMA-2. Total batch size is set to $512$. Batch size is set to $256$ for training 130M LLAMA-2 and is set to $128$ for training 350M LLAMA-2. Dtype is bfloat16. The initial learning rate is $0.001$ and weight decay is $0.0$.

**Settings on training GPT-2 on OWT.** We run Adafactor/AdamW with 2000 warmup steps. Total batch size is set to $480$. Batch size is set to $24$ for training 124M GPT-2. Dtype is bfloat16. The initial learning rate is $0.0006$ and weight decay is $0.1$.

**Settings on training ResNet50 on ImageNet-1k.** We run SGDM (Qian, 1999)/AdamW/Adafactor for 100 epochs with a linear warmup at the first 10 epochs. Minibatch size is set to $512$. For SGDM, we set momentum decay $\beta$ to $0.9$, the initial learning rate to $0.1$, and the weight decay to $0.0005$. For AdamW/Adafactor, we set the initial learning rate to $0.001$, and the weight decay to $0.05$. We adopt the cosine learning rate schedule. Data augmentation follows the configuration for training ResNet50 in (Zhou et al., 2023). We utilize PyTorch's native Automatic Mixed Precision (AMP) functionality (torch.cuda.amp) for training.

**Settings on training ViT-Base/16 on ImageNet-1k.** We run Adafactor/AdamW for 150 epochs with a linear warmup at the first 10 epochs. Minibatch size is set to $512$. The initial learning rate is $0.001$ and weight decay is $0.05$. We use the cosine learning rate schedule. Data augmentation follows the configuration for training ViT-Base/16 in (Zhou et al., 2023), excluding repeated augmentation. We utilize PyTorch's native Automatic Mixed Precision (AMP) functionality (torch.cuda.amp) for training.

**Settings on fine-tuning 7B models.** Dtype is bfloat16. Training epochs is set to 3. Batch size is set to 2 and the gradient accumulation steps is 32. The initial learning rate is $0.00003$ and weight decay is $0.0$. We set warmup_ratio $= 0.3$ and adopt cosine learning rate decay.

**Settings on running the GLUE benchmark.** We set batch_size $=$ auto and num_fewshot $= 5$.

