# OpenReview forum: "2D Quantization for Ultra‑low‑bit Optimizers"
_ICLR.cc/2026/Conference — ICLR 2026 Conference Withdrawn Submission_

### Official Review · Reviewer_5Kms · 2025-10-25

**Soundness:** 2
**Presentation:** 2
**Contribution:** 2
**Rating:** 4
**Confidence:** 5

**Summary:**

This paper proposes a 2D polar quantization scheme for optimizer states that enables ultra-low-bit AdamW/Adafactor (1.5 or 2.0 bits per parameter). The key idea is to jointly quantize pairs of values using a fixed but structured codebook by leveraging the geometry observed in optimizer states. Experiments on LLM pretraining (GPT-2-124M; LLaMA-2-130M/350M), LLM fine-tuning (LLaMA-2-7B, Qwen-2.5-7B), and vision pretraining are conducted to demonstrate the effectiveness of this scheme.

**Strengths:**

- The 2D quantization scheme applied to optimizer states is novel to me.
- Most of the writing is clear.

**Weaknesses:**

- The design of the quantization maps uses magic numbers and lacks justification.
- Section 5.2 lacks further discussion. What is the intuition behind using \alpha? More specifically, why do we need it and how should we scale it? The current manuscript provides little intuition on this, which fails to convince me.
- Writing: Section 4, particularly 4.2, is too technical and not well written.
- In my view, the experiments are not very solid overall. The total compute used in this project appear to be relatively limited.

**Questions:**

- In the 2D quantization scheme, two consecutive elements are treated as one entity. In this simple grouping method, as the authors say, the data can exhibit a near-quasi-Gaussian circular distribution. Could you provide more clarification and justification for your this claim?
- Regarding memory overhead: Although Table 3 shows good memory reduction (2x from 4-bit to 2-bit), other tables such as Tables 2 and 4 do not exhibit much improvement compared to 4-bit, indicating possible memory overheads (e.g., scale factors) in the 2-bit implementation. Could you clarify this in detail?
- I am curious about how you implement Equation (7) in practice.

---

### Official Review · Reviewer_KMxQ · 2025-10-29

**Soundness:** 3
**Presentation:** 2
**Contribution:** 3
**Rating:** 6
**Confidence:** 3

**Summary:**

This paper introduces a 2D polar quantization framework for low-bit storage of optimizer states (such as AdamW and Adafactor) in neural network training. The core idea is to exploit the quasi-Gaussian, circularly symmetric distribution of optimizer states by mapping high-precision values into a structured, 2D polar codebook representation.  The method is validated with static quantization experiments and a suite of pretraining and fine-tuning benchmarks across language and vision models.

**Strengths:**

I think this idea is easy to use and reasonable, and the experiments demonstrate its effectiveness. However, the authors inserted a large number of mathematical derivations in the methods section, which makes the paper difficult to follow.

**Weaknesses:**

1. Why is quantization applied only to the optimizer states, but not to activations or weights (I think weights are also quasi-Gaussian)? What issues would arise if the same quantization were applied to neurons and weights? What are the fundamental differences in data characteristics among activations, weights, and optimizer states?

2. In Figure 3, some curves have not yet converged—what are the final convergence results? In addition, the beginnings of these curves are truncated, so how do they perform at the start?

3. If Figure 3 were plotted against the number of training steps instead of training time, how would the results compare? This relates to the efficiency of data utilization during training.

4. I suggest the authors provide pseudocode (in algorithmic or Python form) for the quantize and dequantize processes, so that readers can verify their own implementations, since many readers in this field may care more about the engineering details.

5. I also recommend that the authors highlight the best results in the tables using boldface or another clear marker. It is difficult to compare all the numbers one by one, and some metrics are better when higher (e.g., accuracy), while others are better when lower (e.g., memory usage).

**Questions:**

1. Why does the performance of C4 Adafactor in Figure 3 become worse as the bit width increases, which is the opposite of other experiments?
weaknesses

2. Perhaps I missed this: how is the codebook radius chosen?

---

### Official Review · Reviewer_waGS · 2025-10-30

**Soundness:** 3
**Presentation:** 3
**Contribution:** 2
**Rating:** 4
**Confidence:** 5

**Summary:**

This paper explores techniques to reduce the precision of optimizer states to as low as 2 or 1.5 bits for memory savings when training deep learning models. Specifically, it proposes a 2D polar quantizer to quantize the first and second moments of the AdamW and Adafactor optimizers. The method is evaluated on various deep learning applications and benchmarks, demonstrating improved performance under ultra-low precision settings.

**Strengths:**

1.	The paper is reasonably well written.

2.	The evaluation covers a wide range of applications.

3.	The results demonstrating good performance under ultra-low precision settings.

**Weaknesses:**

1.	The proposed method appears to be a modification of standard vector quantization, by mapping vectors onto a unit circle. However, the paper provides no strong justification or empirical evidence for why this mapping is important, aside from mentioning that the states follow a “quasi-Gaussian distribution.” No ablation studies are provided to demonstrate the significance of this specific mapping.

2.	The quantization error analysis and benchmark comparisons are questionable. The paper uses a relatively small quantization group size of 64, but in Table 1, which compares quantization errors, it does not specify the group sizes used for other methods. Similarly, in Tables 2–4 (benchmark results), the group sizes for the baselines are not provided.

3.	The experimental results (Tables 2–4) lack comparisons with state-of-the-art quantizer baselines. The paper only includes a single 2/4-bit baseline without specifying the experimental conditions. It is important to compare against other quantizers mentioned in the paper to properly assess the advantages of the proposed method.

4.	The evaluated language models are relatively small in scale.

5.	There is a lack of ablation studies on normalization, group size, codebook size, and the introduced hyperparameter α.

**Questions:**

(Following the weaknesses)

1.	What quantization group sizes were used in the baselines for both the quantization error comparison and the final benchmark results?

2.	Which component of the technique has the greatest impact on performance — the polar coordinate mapping, group size, or codebook size?

3.	What is the computational overhead of the proposed quantizer, for example, quantization/dequantization and normalization?

---

### Official Review · Reviewer_J381 · 2025-11-01

**Soundness:** 1
**Presentation:** 1
**Contribution:** 3
**Rating:** 2
**Confidence:** 3

**Summary:**

This paper proposes a 2D quantization method for compressing optimizer states (momentum and second moments) to 1.5 and 2 bits, enabling memory-efficient training. The key innovation is to quantize pairs of consecutive parameters jointly in polar coordinates rather than independently, exploiting the claimed circular symmetry in optimizer state distributions. The method is evaluated on AdamW and Adafactor across language modeling (GPT-2, LLaMA-2) and vision tasks (ViT, ResNet-50), demonstrating comparable performance to full-precision baselines while achieving up to 4.6x memory reduction.

**Strengths:**

- The experimental coverage is broad, spanning both language modeling (GPT-2, LLaMA-2 pretraining and fine-tuning) and vision tasks (ViT, ResNet-50), with consistent memory savings demonstrated across these diverse domains.
- The idea of leveraging 2D geometry (polar coordinates) for quantization is novel and interesting.
- The paper correctly identifies that unsigned (second moment) and signed (first moment) states require different treatment, and proposes reasonable quantization strategies for each case.
- The paper addresses an important problem---reducing optimizer memory footprint during training---and achieves meaningful compression ratios (1.5-2 bits) that go beyond prior work's 4-bit limit. The experimental results are promising, showing stable training and competitive performance across diverse benchmarks.

**Weaknesses:**

- The current theory section is very sloppy and lacks rigor.
    - Theorem 3, as it stands, is incorrectly stated. Theorem 3 states

        > Suppose $f: \mathbb{R} \rightarrow \mathbb{C}$ is defined as $f(t) = e^{i\theta} + e^{it\theta}$, where $\theta$ is irrational. Then $f(\mathbb{R})$ is dense in $\{z \in \mathbb{C} : |z| \leq 2\}$.

        This is demonstrably false. The function $f(t)$ represents the sum of a fixed point $e^{i\theta}$ on the unit circle and a point $e^{it\theta}$ that traces the unit circle. The image of $f$ is therefore a circle of radius 1 centered at $e^{i\theta}$, which covers only points exactly 1 unit away from $e^{i\theta}$---not the entire disk of radius 2. I believe the authors may have intended to define $f(t) = e^{it} + e^{i t \theta}$, where $\theta$ is irrational.

    - Even if the theorem statement were corrected (as suggested above), It's not immediately apparent to me that this corrected version is a valid claim either (specifically regarding $f(\mathbb{R})$ being dense in the disk). A rigorous proof is needed.
    - The derivation leading to Theorem 3 lacks rigor. The discussion preceding the theorem relies on informal arguments and does not constitute a mathematical proof. The theoretical framework appears heavily based on the π-Quant paper (Tian et al., 2025) and its Theorem 3.1. However, π-Quant's Theorem 3.1 also lacks a rigorous proof.
        - If this work builds on π-Quant's theory, this should be explicitly acknowledged, and ideally, rigorous proofs should be provided where the original work fell short.
        - The paper mentions π-Quant only briefly ("due to theoretical and implementation limits, [π-Quant] only reduces AdamW bitwidth to 3.32 bits").
    - Theorem 2, as it is currently stated, is identical to Theorem 1, but is labeled differently. This error appears both in the main text and in the appendix. This appears to be a copy-paste error and needs correction.

- The paper's motivation hinges on the claim that optimizer states exhibit "strong circular symmetry" when viewed as 2D vectors. In the Introduction section, the authors state:

    > As we empirically demonstrate in Appendix D, optimizer states consistently exhibit a quasi-Gaussian distribution with strong circular symmetry.

    However, the two figures provided in Appendix D do not substantiate this claim:
    - Figure 5 shows only 1D histograms for the two different optimizer states, justifying different designs for signed vs. unsigned states but revealing nothing about 2D geometry.
    - Figure 6 shows 1D distributions after quantization/dequantization, demonstrating the method's effectiveness as a result but not the premise.
    - What is missing: Any kind of 2D visualization (e.g., scatter plots of `(x[2i], x[2i+1])` pairs) from real optimizer states to directly demonstrate circular symmetry. Without this, the foundational motivation for the 2D quantization approach is unsubstantiated.

    Provided that key parts of the theory in this paper build on the assumption of circular symmetry, this empirical validation is critical.

- Some of the key implementation details are unclear.
    - The practical implementation of the quantization method is not clearly described. Section 4.2 discusses theoretical foundations but never clearly explains the practical quantization implementation. Does the method use nearest-neighbor search, or does it leverage the theory to construct a fast quantization algorithm? The connection between the theory and the practical implementation is opaque.
    - The specific radii values (e.g., $R=\{0.40\}$ for 1.5-bit signed, $R=\{0.14, 0.53\}$ for 2.0-bit signed) appear arbitrary. The paper mentions "placing codebook magnitudes near the median is beneficial" but provides no principled derivation or ablation study justifying these choices.

- Presentation/experimental clarity issues
    - The authors mention (see quote below) that several parameter groups are intentionally kept in full precision (see quote below). How were these handled in the baselines? Can the authors clearly clarify, for each and every baseline, whether these same selective quantization strategies were applied?
        > Conversely, several parameter groups are intentionally kept in full precision and are thus excluded from $S_{Q}$. For Large Language Models (LLMs), these non-quantized parts notably include the embedding layer parameters, the bias vectors within linear layers, and all parameters within normalization layers (e.g., LayerNorm, RMSNorm).
    - Given the apparent relationship to π-Quant, a direct experimental comparison would strengthen the paper. Table 4 also lacks AdamW results, breaking the pattern established in other tables.
    - Table 4 (fine-tuning) reports "total memory cost (TMC)" while all other tables report "GPU memory usage of optimizer states (MC)." This inconsistency makes cross-table comparisons difficult and raises questions about what is being measured.
    - The "Original" baseline in Table 4 is undefined (which optimizer? which bitwidth?).

**Questions:**

1. Can you provide a corrected statement and rigorous proof of Theorem 3? How does your theoretical contribution relate to π-Quant's Theorem 3.1, and what are the key differences or improvements?
2. For the empirical demonstration, can you provide 2D scatter plots of `(x[2i], x[2i+1])` from real optimizer states (or any other 2D visualization on `(x[2i], x[2i+1])`) to directly demonstrate the circular symmetry? Under what conditions does this symmetry emerge, and does it hold universally or only for certain layer types/training stages?
3. Does your method use nearest-neighbor search or does it exploit the structure described in Section 4.2 for faster quantization? If the latter, please provide a clear description of the algorithm in the paper (in addition to addressing the theoretical concerns).
4. How were the specific radius values chosen? Was this through grid search, or is there a principled method (e.g., based on quantiles of the distribution)?
5. In Table 4, going from 8.0-bit to 1.5-bit Adafactor reduces TMC by only ~5GB (54GB -> 49GB for LLaMA2-7B, 63GB -> 58GB for Qwen2.5-7B). For a 5.3x quantization improvement, the memory reduction seems disproportionately small. Can you explain why this is the case?

---

### Note · Authors · 2025-11-14

I have read and agree with the venue's withdrawal policy on behalf of myself and my co-authors.